# Genomic analysis finds no evidence of canonical eukaryotic DNA processing complexes in a free-living protist

Dayana E. Salas-Leiva [1,2✉], Eelco C. Tromer [2,3], Bruce A. Curtis [1], Jon Jerlström-Hultqvist [1], Martin Kolisko [4], Zhenzhen Yi [5], Joan S. Salas-Leiva [6], Lucie Gallot-Lavallée [1], Shelby K. Williams [1], Geert J. P. L. Kops [7], John M. Archibald [1], Alastair G. B. Simpson [8] & Andrew J. Roger [1✉]

Cells replicate and segregate their DNA with precision. Previous studies showed that these regulated cell-cycle processes were present in the last eukaryotic common ancestor and that their core molecular parts are conserved across eukaryotes. However, some metamonad parasites have secondarily lost components of the DNA processing and segregation apparatuses. To clarify the evolutionary history of these systems in these unusual eukaryotes, we generated a genome assembly for the free-living metamonad *Carpediemonas membranifera* and carried out a comparative genomics analysis. Here, we show that parasitic and free-living metamonads harbor an incomplete set of proteins for processing and segregating DNA. Unexpectedly, *Carpediemonas* species are further streamlined, lacking the origin recognition complex, Cdc6 and most structural kinetochore subunits. *Carpediemonas* species are thus the first known eukaryotes that appear to lack this suite of conserved complexes, suggesting that they likely rely on yet-to-be-discovered or alternative mechanisms to carry out these fundamental processes.

[1] Institute for Comparative Genomics (ICG), Department of Biochemistry and Molecular Biology, Dalhousie University, Halifax, NS B3H 4R2, Canada. [2] Department of Biochemistry, University of Cambridge, Cambridge, United Kingdom. [3] Groningen Biomolecular Sciences and Biotechnology Institute, University of Groningen, Groningen, Netherlands. [4] Institute of Parasitology, Biology Centre, Czech Acad. Sci, České Budějovice, Czech Republic. [5] Guangzhou Key Laboratory of Subtropical Biodiversity and Biomonitoring, School of Life Science, South China Normal University, Guangzhou 510631, China. [6] CONACyT-Centro de Investigación en Materiales Avanzados, Departamento de medio ambiente y energía, Miguel de Cervantes 120, Complejo Industrial Chihuahua, 31136 Chihuahua, Chih., México. [7] Oncode Institute, Hubrecht Institute – KNAW (Royal Netherlands Academy of Arts and Sciences) and University Medical Centre Utrecht, Utrecht, The Netherlands. [8] Institute for Comparative Genomics (ICG), Department of Biology, Dalhousie University, Halifax, NS B3H 4R2, Canada. ✉email: Dayana.Salas@dal.ca; Andrew.Roger@dal.ca

DNA replication, repair, and segregation are critically important and conserved processes in eukaryotes that have been intensively studied in model organisms[1]. The initial step of DNA replication is accomplished by the replisome, a set of highly conserved proteins that is tightly regulated to minimize mutations[2]. The replisome relies on the interactions between cis-acting DNA sequences and trans-acting factors that serve to separate the template and promote RNA-primed DNA synthesis. This occurs by the orderly assembly of the origin recognition (ORC), the pre-replicative (pre-RC), pre-initiation (pre-IC) and replication progression (RPC) complexes[3–6]. The synthesis of DNA usually encounters disruptive obstacles as replication proceeds and can be rescued either through template switching via trans-lesion or recombination-dependent synthesis. Trans-lesion synthesis uses replicative and non-replicative DNA polymerases to bypass the lesion through multiple strategies that incorporate nucleotides opposite to it, while recombination-dependent synthesis uses nonhomologous or homologous templates for repair (reviewed in refs. [7,8]). Recombination-dependent synthesis occurs in response to single- or double-strand DNA breakage[8–10]. Other repair mechanisms occur throughout the cell cycle, fixing single-strand issues through base excision (BER), nucleotide excision (NER), or mismatch (MMR) repair, but they may also be employed during replication depending on the source of the damage. All of the repair processes are overseen by multiple regulation checkpoints that permit or stall DNA replication and the progression of the cell cycle. During M-phase the replicated DNA has to form attachments with the microtubule-based spindle apparatus via kinetochores (KTs), large multi-subunit complexes built upon centromeric chromatin[11]. Unattached KTs catalyze the formation of a soluble inhibitor of the cell cycle, preventing precocious chromosome segregation, a phenomenon known as the spindle assembly checkpoint (SAC)[11]. Failure to pass any of these checkpoints (e.g., G1/S, S, G2/M, and SAC checkpoints reviewed in refs. [11–13]) leads to genome instability and may result in cell death.

To investigate the diversity of DNA replication, repair, and segregation processes, we conducted a eukaryote-wide comparative genomics analysis with a special focus on metamonads, a major protist lineage comprised of parasitic and free-living anaerobes. Parasitic metamonads such as *Giardia intestinalis* and *Trichomonas vaginalis* are highly divergent from model system eukaryotes, exhibit a diversity of cell division mechanisms (e.g., closed/semi-open mitosis), possess metabolically reduced mitosomes or hydrogenosomes instead of mitochondria, and lack several canonical eukaryotic features on the molecular and genomic-level[14–16]. Indeed, recent studies show that metamonad parasites have secondarily lost parts of the ancestral DNA replication and segregation apparatuses[17,18]. Furthermore, metamonad

proteins are often highly divergent compared to other eukaryotic orthologs, indicating a high substitution rate in these organisms that is suggestive of error-prone replication and/or DNA repair[19]. Yet, it is unclear whether the divergent nature of proteins studied in metamonads is the result from the host-associated lifestyle or is a more ancient feature of Metamonada. To increase the representation of free-living metamonads in our analyses, we have generated a high-quality draft genome assembly of *Carpediemonas membranifera*, a flagellate isolated from hypoxic marine sediments.

In this work, we show that many systems for DNA replication, repair, segregation, and cell cycle control are ancestral to eukaryotes and highly conserved. However, metamonads have secondarily lost a large number of components. Most remarkably, the free-living *Carpediemonas* species appear to be further reduced, lacking evidence of key proteins from the replisome and cell cycle checkpoints (i.e., including several from the KT and DNA repair pathways). We propose a hypothesis on how DNA replication may be achieved in these organisms.

## Results

**The *C. membranifera* genome assembly is complete.** Our assembly for *C. membranifera* is very contiguous (Table 1) and has deep read coverage (i.e., median coverage of 150× with short reads and 83× with long reads), with estimated genome completeness of 99.27% based on the Merqury[20] method. 97.6% of transcripts mapped to the genome along their full length with an identity of ≥95% while a further 2.04% mapped with an identity between 90−95%. The *C. membranifera* genome size is small compared to that of other free-living metamonads (e.g., *Kipferlia bialata*), has a high GC content (57.1%), and is among the most contiguous assemblies of any metamonads included in our study. The high contiguity of the assembly is underscored by the large number of transcripts mapped to single contigs (90.2%), and since the proteins encoded by transcripts were consistently found in the predicted proteome, the latter is also considered to be of high quality. We also conducted BUSCO analyses, with the foreknowledge that genomic streamlining typical in Metamonada has led to the loss of many conserved proteins[15,16]. Our analyses show that previously completed metamonad genomes only encoded between 60 to 91% of the BUSCO proteins, while *C. membranifera* encodes a relatively high number of 89% of BUSCO proteins (Table 1, Supplementary Information, and Supplementary Data 1). In any case, our coverage estimates for the *C. membranifera* genome for short and long-read sequencing technologies are substantially greater than those found to be sufficient to capture genic regions that otherwise would have been missed (i.e., coverage >52× for long reads and >60× for short

**Table 1 Summary statistics of nuclear genomes of Metamonada species.**

| Taxa | Genome size (Mb) | Contigs | N50 (Kb) | GC (%) | Predicted proteins | BUSCO (genes) | BUSCO (%) |
|---|---|---|---|---|---|---|---|
| *Trichomonas vaginalis* | 176.4 | 64,764 | 27.2 | 32.9 | 95,606 | 223 | 91 |
| *Monocercomonoides exilis* | 74.7 | 2095 | 71.4 | 37.4 | 16,780 | 224 | 91 |
| *Carpediemonas membranifera* | 24.2 | 69 | 905.8 | 57.1 | 8300 | 217 | 89 |
| *Carpediemonas frisia* | 12.6 | 3232 | 9.5 | 58.6 | 5695 | 184 | 75 |
| *Kipferlia bialata* | 51.0 | 11,563 | 10.5 | 47.8 | 17,389 | 207 | 84 |
| *Spironucleus salmonicida* | 12.9 | 233 | 150.8 | 33.5 | 8354 | 152 | 62 |
| *Trepomonas sp. PC1** | | | | | 7980 | 147 | 60 |
| *Giardia intestinalis* A-50803 | 12.8 | 211 | 2762.4 | 49.2 | 5901 | 168 | 69 |
| *Giardia intestinalis* B-50581 | 11.0 | 2931 | 36.6 | 46.9 | 4470 | 169 | 69 |
| *Giardia muris* | 9.8 | 59 | 2398.6 | 54.7 | 4936 | 173 | 71 |

All the statistics were recalculated with Quast v5.0.2[97] for completion as not all of these were originally reported, and the BUSCO reference protein set corresponds to a maximum of 245 proteins.
*Transcriptome data only.

paired-end reads, see ref. [21]). All these various data indicate that the draft genome of *C. membranifera* is nearly complete; if any genomic regions are missing, they are likely confined to difficult-to-sequence repetitive regions such as telomeres and centromeres.

Note that a previous study conducted a metagenomic assembly of a related species, *Carpediemonas frisia*, together with its associated prokaryotic microbiota[22]. For completeness, we have included these data in our comparative genomic analyses (Table 1, Supplementary Information), although we note that the *C. frisia* metagenomic bin is based on only short-read data and might be partial.

To generate an up-to-date phylogenetic framework for our comparative genomic analyses, we conducted a phylogenomic analysis with a broad sampling of the Metamonada and selected outgroup taxa. The resulting topology (Supplementary Fig. 1) was highly supported and recovered the same within-group metamonad and fornicate relationships as previous analyses (see refs. [22,23]). Specifically, the two *Carpediemonas* species form a well-supported clade that emerges from the deepest division within Fornicata (i.e., the clade comprised of diplomonads, retortamonads, and *Carpediemonas*-like organisms (CLOs)). This analysis also demonstrates that, with the exception of *Trimastix* and *Paratrimastix*, metamonads form very long branches on the tree (i.e., ~1.5-fold to threefold longer than outgroup branches), with the diplomonad sequences being the most divergent.

**Streamlining of the DNA replication apparatus in metamonads.** The first step in the replication of DNA is the assembly of ORC which serves to nucleate the pre-RC formation. The initiator protein Orc1first binds an origin of replication, followed by the recruitment of Orc 2–6 proteins, which associate with chromatin[24]. As the cell transitions to the G1 phase, the initiator Cdc6 binds to the ORC, forming a checkpoint control[25]. Cdt1 then joins Cdc6, promoting the loading of the replicative helicase MCM forming the pre-RC, a complex that remains inactive until the onset of the S-phase when the "firing" factors are recruited to convert the pre-RC into the pre-IC[3–5]. Additional factors join to form the RPC to stimulate replication elongation[26]. The precise replisome protein complement varies somewhat between different eukaryotes, suggesting that some of these proteins may not be essential or could indicate some degree of functional impairment. However, metamonads show more variation in ORC, pre-RC, and replicative polymerases (Fig. 1, Supplementary Information, and Supplementary Data 2). The presence-absence of ORC and Cdc6 proteins is notably patchy across Metamonada, but our workflow retrieved previously unreported Orc5 orthologs in *T. vaginalis* and *Monocercomonoides exilis* and additional members of the Orc1/Cdc6 protein family to those previously identified in *Giardia* (Supplementary Data 2 and Supplementary Fig. 2). Our detection of these homologs was facilitated by the broad amino acid sequence diversity encompassed by the taxa-enriched HMMs (Hidden Markov Models) that increased the sensitivity of our searches, enabling retrieval of these highly divergent homologs. Strikingly, whereas most metamonads retain up to two paralogs of the core protein family Orc1/Cdc6 (here called Orc1 and Orc1/Cdc6-like as their precise assignment is difficult, see Supplementary Fig. 3), plus some orthologs of Orc 2–6, all these proteins are absent in *C. membranifera* and *C. frisia* (Fig. 1 and Supplementary Data 2). The lack of all of these proteins in a eukaryote is unexpected, since their absence is expected to make the genome prone to double strand breaks (DSBs) and impair DNA replication, as well as interfere with other non-replicative processes[27]. To rule out false negatives, we conducted further analyses using metamonad-specific HMMs, various other profile-based search strategies (Supplementary Information and

Supplementary Data 3), tBLASTn v.2.7.1[28] searches (i.e., on the genome assembly and unassembled long reads), and applied HMMER v3.1b2[29] searches on six-frame assembly translations. These additional methods were sufficiently sensitive to identify these proteins in all nuclear genomes we examined, with the exception of the *Carpediemonas* species and the highly reduced, endosymbiotically-derived nucleomorphs of cryptophytes and chlorarachniophytes (Fig. 1, Supplementary Information, and Supplementary Fig. 4). *Carpediemonas* species are, therefore, the only known eukaryotes to lack ORC and Cdc6.

**DNA damage repair systems have undergone several modifications.** DNA repair occurs continuously during the cell cycle depending on the type or specificity of the lesion. Among the currently known mechanisms are BER, NER, MMR, and DSB repair, with the latter conducted by either homologous recombination (HR), canonical nonhomologous end joining (NHEJ), or alternative end joining (a-EJ)[7,13]. MMR can be coupled directly to replication or play a role in HR. MMR, BER, and NER are present in all studied taxa (Supplementary Data 2), although our analyses indicate that damage sensing and downstream functions in NER seem to be modified in the metamonad taxa Parabasalia and Fornicata due to the absence of the XPG and XPC sensor proteins.

DSBs are very dangerous for cells and can occur as a result of damaging agents or from self-inflicted cuts during DNA repair and meiosis. NHEJ requires the heterodimer Ku70–Ku80 to recruit the catalytic kinase DNA-PKcs and accessory proteins. Metamonads lack all of these proteins, as do a number of other eukaryotes investigated here and in ref. [30]. The a-EJ system seems to be fully present in metamonads like *C. membranifera*, partial in others, and absent in parasitic diplomonads. NHEJ is thought to be the predominant mechanism for repairing DSBs in eukaryotes, but since our analyses indicate this pathway is absent in metamonads and a-EJ is highly mutagenic[7], the HR pathway is likely to be essential for DSB repair in most metamonads. Repair by the HR system occurs through multiple sub-pathways that are influenced by the extent of the similarity of the DNA template or its flanking sequences to the sequences near the break. HR complexes are recruited during DNA replication and transcription and utilize DNA, transcript-RNA, or newly synthesized transcript-cDNA as a homologous template[10,31–34]. These complexes are formed by recombinases from the RecA/Rad51 family that interact with members of the Rad52 family and chromatin remodeling factors of the Snf2/Swi2 subfamily. Although the recombinases Rad51A-D are all present in most eukaryotes, we found a patchy distribution in metamonads (Supplementary Data 2 and Supplementary Fig. 5). All examined Fornicata have lost the major recombinase Rad51A and have two paralogs of the meiosis-specific recombinase Dmc1, as first noted in *Giardia intestinalis*[35]. Dmc1 has been reported to provide high stability to recombination due to strong D-loop resistance to strand dissociation[36]. The recombination mediator Rad52 is present in most metamonads but Rad59 or Rad54 are not. Metamonads have no components of an ISWI remodeling complex yet retain a reduced INO80 complex. Therefore, replication fork progression and HR are likely to occur under the assistance of INO80 alone. HR requires endonucleases and exonucleases, and our searches for proteins additional to those from the MMR pathway revealed a gene expansion of the Flap proteins from the Rad2/XPG family in some metamonads. We also found proteins of the Pif1 helicase family that encompasses homologs that resolve R-loop structures, unwind DNA–RNA hybrids, and assists in fork progression in regular replication and HR[37,38]. Phylogenetic analysis reveals that although

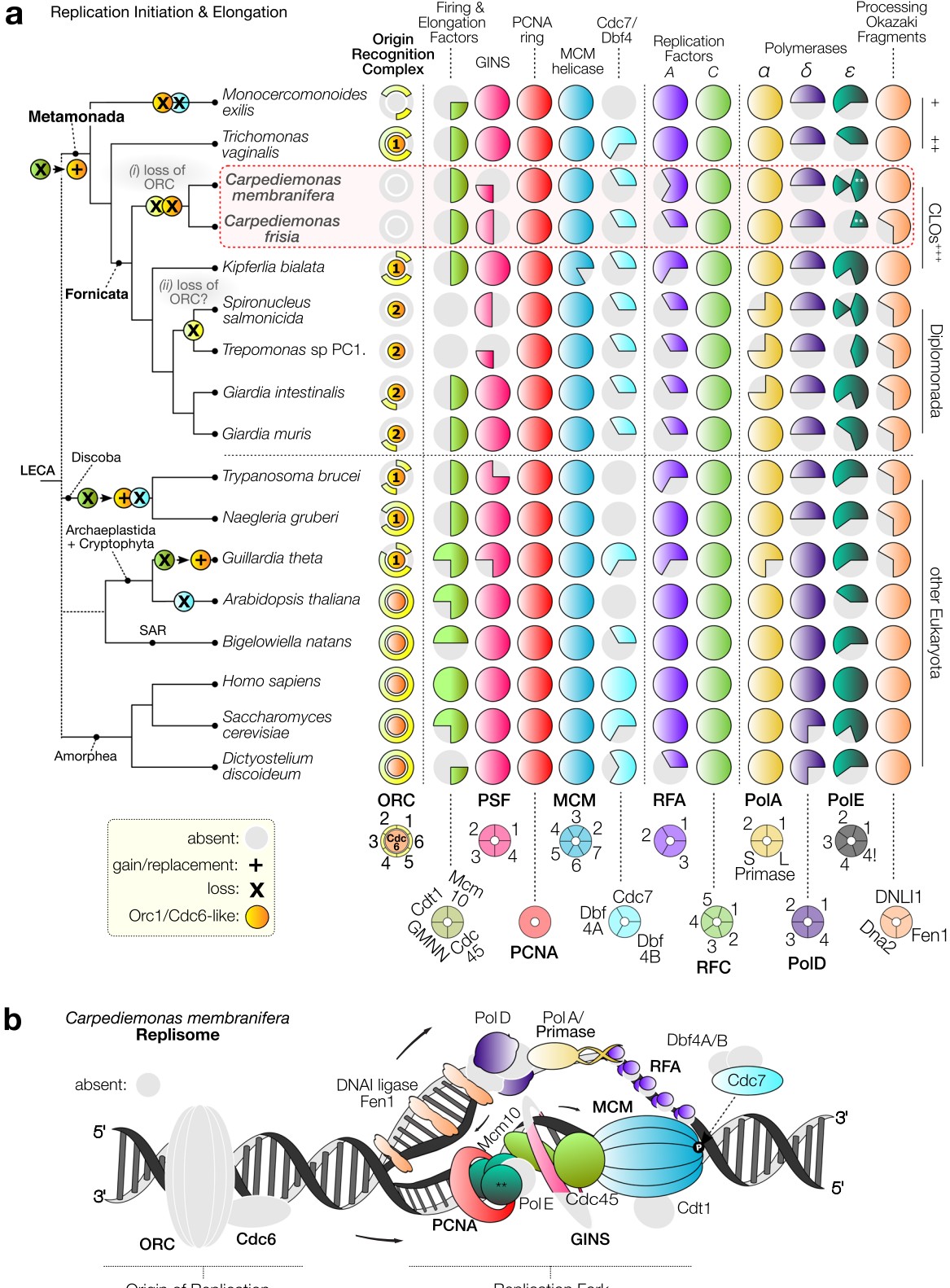

**a** Replication Initiation & Elongation

**b** *Carpediemonas membranifera* **Replisome**

*Carpediemonas* species have orthologs that branch within a metamonad group in the main Pif1 clade (Fig. 2), they also possess a highly divergent clade of Pif1-like proteins. Each *Carpediemonas* species has multiple copies of Pif1-like proteins that have independently duplicated within each species; these may point to the de novo emergence of specialized functions in HR and DNA replication for these proteins. Metamonads appear

capable of using all the HR sub-pathways (e.g., classical DSB repair, single-strand annealing, and break-induced replication), but these are modified (Supplementary Data 2 and Supplementary Fig. 5). Overall, the presence-absence patterns of the orthologs involved in DSB repair in Fornicata point to the existence of a highly specialized HR pathway which is presumably not only essential for the cell cycle of metamonads

**Fig. 1 The distribution of core molecular systems in the replisome and DNA repair across eukaryotic diversity. a** A schematic global eukaryote phylogeny is shown on the left with the phylogeny of the major metamonad lineages based on our phylogenomic analysis (Supplementary Fig. 1). The classification of the major lineages is indicated on the right. Reduction of the replication machinery and loss of the Orc1–6 subunits are observed in metamonad lineages, including the unexpected loss of the highly conserved ORC complex and Cdc6 in *Carpediemonas*. Most metamonad Orc1 and Cdc6 homologs were conservatively named as "Orc1/Cdc6-like" as they are very divergent, do not have the typical domain architecture and, in phylogenetic reconstructions, they form clades separate from the main eukaryotic groups, preventing confident orthology assignments (Supplementary Fig. 3). Numbers within circles represent the number of gene copies and are only presented for ORC components; each column is color-coded by complex or protein group and the same color coding is used to depict proteins in panel **b**; circles on branches of the tree represent the loss/gain/replacement of the proteins/ complexes represented by the circle coloring scheme. Cdc6 and Orc1/Cdc6-like proteins are represented with a darker yellow for which the number of homologs is shown, additional information in Supplementary Data 2, 5, and 6. The polymerase epsilon (ε) is composed of four subunits, but we included the interacting protein Chrac1 (depicted as "4!" in the figure) as its HMM retrieves the polymerase delta subunit Dbp3 from *S. cerevisiae*. *Firing and elongation factors, **Protein fusion between the catalytic subunit and subunit 2 of DNA polymerase ε. +Preaxostyla, ++Parabasalida, +++*Carpediemonas*- like organisms. **b** The predicted *Carpediemonas* replisome components (colored) overlaid on features of a typical eukaryotic replisome. Origin recognition (ORC), Cdc6, and replication progression (RPC) complexes are depicted. Shapes in gray indicate the absence of typical eukaryotic replisome proteins in *C. membranifera*.

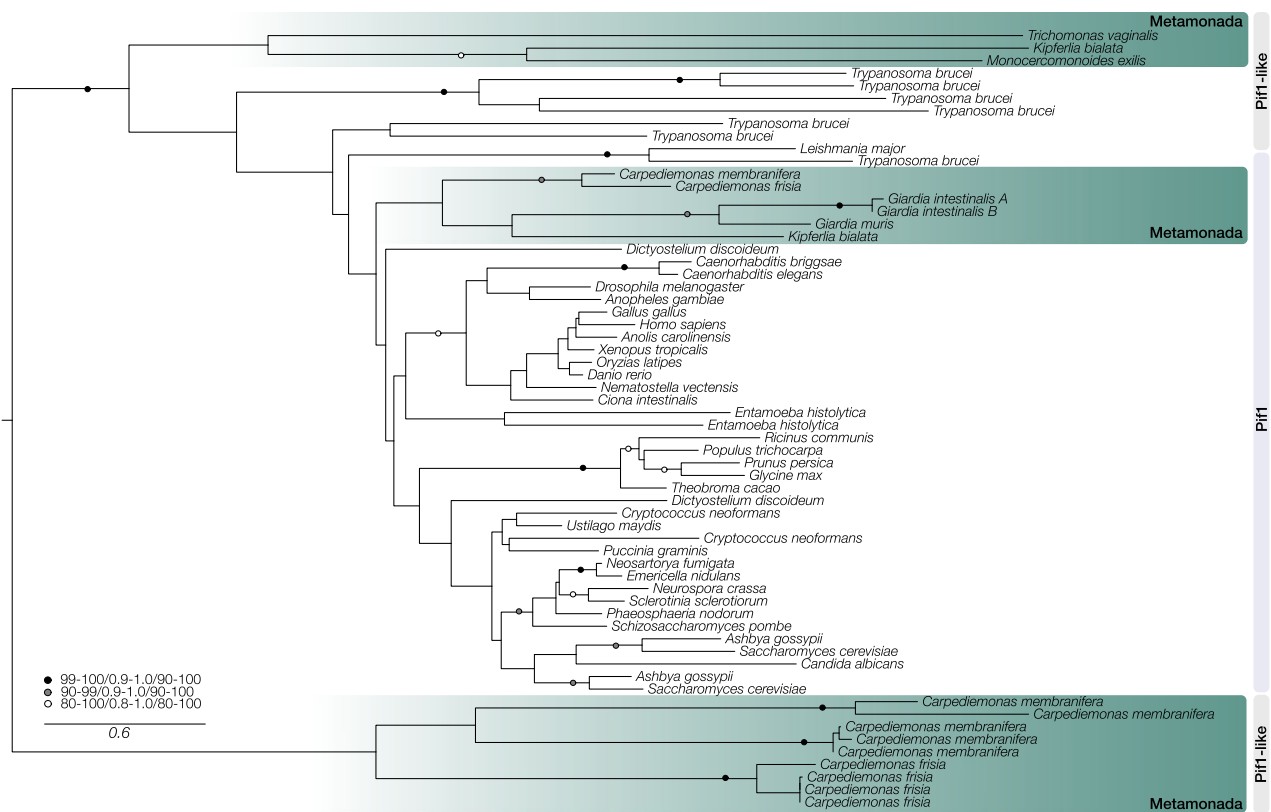

**Fig. 2 Pif1 protein family expansion. Pif1 helicase family tree.** Three clades are highlighted: at the top, a Pif1-like clade encompassing some metamonads and at the bottom a *Carpediemonas*-specific Pif1-like clade. The third clade shows the typical Pif1 orthologs encompassing fornicates. The maximum-likelihood tree was inferred under the LG + PMSF(C60) + F + Γ model using 100 bootstraps based on an alignment length of 265 sites. The tree was midpoint-rooted and the support values on the branches correspond to SH-aLRT/aBayes/standard bootstrap (only values above 80/0.8/80 are depicted). The scale bar shows the inferred number of amino acid substitutions per site.

but is also likely the major pathway for replication-related DNA repair and recombination.

**Modified DSB damage response checkpoints in metamonads.** Checkpoints constitute a cascade of signaling events that delay replication until DNA lesions are resolved[12]. The ATR-Chk1, ATM-Chk2, and DNA-PKcs pathways are activated by the interaction of TopBP1 and the 9-1-1 complex (Rad9-Hus1-Rad1) for DNA repair regulation during replication stress and response to DSBs[39]. The ATR-Chk1 signaling pathway is believed to be the

initial response to ssDNA damage and be responsible for the coupling of DNA replication with mitosis, but when it is defective, the ssDNA is converted into DSBs to activate the ATM-Chk2 pathway. The DNA-PKcs act as sensors of DSBs to promote NHEJ, but we found no homologs of DNA-PKcs in metamonads (Supplementary Fig. 5), which is consistent with the lack of an NHEJ repair pathway in the group. All the checkpoint pathways described are present in humans and yeasts, while the distribution of core checkpoint proteins in the remaining taxa is patchy. Notably, Fornicata lack several of the proteins thought to be needed to activate the signaling kinase cascades and, while

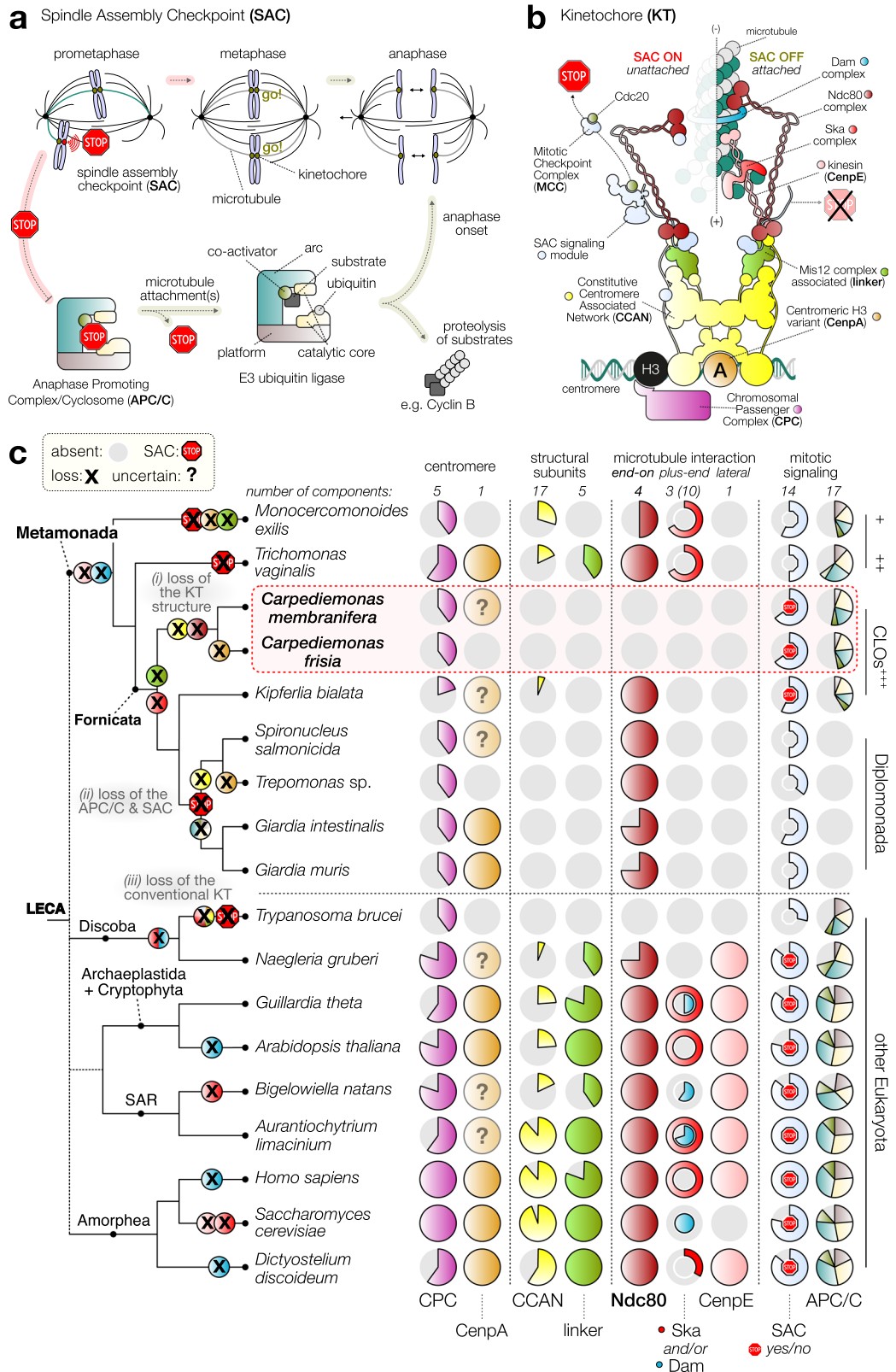

orthologs of ATM or ATR kinases are present in some fornicates, there are no clear orthologs of Chk1 or Chk2 in metamonads except in *M. exilis* (Supplementary Data 2 and Supplementary Fig. 5). *Carpediemonas* species and *K. bialata* contain ATM and ATR but lack Chk1, Chk2, and Rad9. Diplomonads possess none of these proteins. The depletion of Chk1 has been shown to increase the incidence of chromosomal breaks and mis-

segregation[40]. All these absences reinforce the idea that the checkpoint controls in Fornicata are non-canonical.

**Reduction of mitosis and meiosis machinery in metamonads.** Eukaryotes synchronize cell cycle progression with chromosome segregation by a KT-based signaling system called the SAC[41,42]

**Fig. 3 Reduction of ancestral kinetochore network complexity in *Carpediemonas* species. a** Schematic of canonical mitotic cell cycle progression in eukaryotes. During mitosis, each duplicated chromosome attaches to microtubules (MTs) emanating from opposite poles of the spindle apparatus, in order to be segregated into two daughter cells. Kinetochores (KTs) are built upon centromeric DNA to attach microtubules to chromosomes. To prevent precocious chromosome segregation, unattached KTs signal to halt cell cycle progression (STOP), a phenomenon known as the spindle assembly checkpoint (SAC). Once all KTs are correctly attached to spindle MTs and aligned in the middle of the cell (metaphase), the checkpoint is released, and chromosome segregation is initiated (anaphase). **b** Cartoon of the molecular makeup of a single KT unit that was likely present in the last eukaryotic common ancestor (LECA). The cartoon depicts two different kinetochore states: unattached (left), and when bound to a microtubule (right). Colors indicate the various functional complexes and structures present in either attachment state. **c** Reconstruction of the evolution of the kinetochore and mitotic signaling in eukaryotes based on KT protein presence-absence patterns reveals extensive reduction of ancestral complexity and loss of the SAC in most metamonad lineages, including loss of the highly conserved core MT-binding activity of the KT (Ndc80) in *Carpediemonas*. On top/bottom of panel **c**: the number of components per complex and different structural parts of the KT, SAC signaling, and the APC/C. Middle: presence/absence matrix of KT, SAC, and APC/C complexes; one circle per complex, colors correspond to panel **a** and **b**; gray indicates its (partial) loss (for a complete overview see Supplementary Data 4 and Supplementary Fig. 6). The red STOP sign indicates the likely presence of a functional SAC response (see for discussion Supplementary Fig. 6). On the left: cartoon of a phylogenetic tree of metamonad and other selected eukaryotic species with a depiction of the loss events on each branch. Specific loss events of kinetochore and SAC genes in specific lineages are highlighted in color.

that is ancestral to all eukaryotes (Fig. 3a, b). KTs primarily form microtubule attachments through the Ndc80 complex, which is connected through a large network of structural subunits to a histone H3-variant CenpA that is specifically deposited at centromeres[11]. To prevent premature chromosome segregation, unattached KTs catalyze the production of the mitotic checkpoint complex (MCC)[41], a cytosolic inhibitor of the anaphase promoting complex/cyclosome (APC/C), a large multi-subunit E3 ubiquitin ligase that drives progression into anaphase by promoting the proteolysis of its substrates such as various Cyclins[43] (Fig. 3a). Our analysis indicates the reduction of ancestral complexity of these proteins in metamonads (Fig. 3c, Supplementary Data 4, and Supplementary Fig. 6). Surprisingly, such reduction is extensive in *Carpediemonas* species. We found that most structural KT subunits, a microtubule plus-end tracking complex, and all four subunits of the Ndc80 complex are absent (Fig. 3c and Supplementary Fig. 6). None of our additional search strategies led to the identification of Ndc80 complex members, making *Carpediemonas* the only known eukaryotic lineage without it, except for kinetoplastids, which appear to have lost the canonical KT and replaced it by an analogous molecular system, although there is still some controversy about this loss[44,45]. With such widespread absence of KT components it might be possible that *Carpediemonas* underwent a similar replacement process to that of kinetoplastids[44]. We did however find a potential candidate for the centromeric histone H3-variant (CenpA) in *C. membranifera*. CenpA forms the basis of the canonical KT in most eukaryotes[46] (Supplementary Fig. 7). On the other hand, the presence or absence of CenpA is often correlated with the presence/absence of its direct interactor CenpC[18]. Similar to diplomonads, *C. membranifera* lacks CenpC and therefore the molecular network associated with KT assembly on CenpA chromatin may be very different.

Most metamonads encode all MCC components, but diplomonads lost the SAC response and the full APC/C complex[47]. In contrast, only *Carpediemonas* species and *K. bialata* have MCC subunits that contain the conserved short linear motifs to potentially elicit a canonical SAC signal[43,48] (Supplementary Fig. 8). Interestingly, not all of these motifs are present, and most are seemingly degenerate compared to their counterparts in other eukaryotic lineages (Supplementary Fig. 8c). Also, many other SAC-related proteins are conserved, even in diplomonads (e.g., Mad2 and MadBub)[47]. Furthermore, the cyclins in *C. membranifera*, the main target of SAC signaling, have a diverged destruction motif (D-box) in their N-termini (Supplementary Fig. 8c). Collectively, our observations indicate that *Carpediemonas* species could elicit a functional SAC response, but whether this would be KT-based is unclear. Alternatively, SAC-related

genes could have been repurposed for another cellular function(s) as in diplomonads[47]. Given that ORC has been observed to interact with the KT (throughout chromosome condensation and segregation), centrioles, and promotes cytokinesis[27], the lack of Ncd80 and ORC complexes suggest that *Carpediemonas* species possess unconventional cell division systems.

Neither sexual nor parasexual processes have been directly observed in Metamonada[35]. Nonetheless, our surveys confirm the conservation of the key meiotic proteins in metamonads[35], including Hap2 (for plasmogamy) and Gex1 (karyogamy). Unexpectedly, *Carpediemonas* species have homologs from the tmcB family that acts in the cAMP signaling pathway specific for sexual development in *Dictyostelium*[49], and sperm-specific channel subunits (i.e., CatSper α, β, δ, and γ) reported previously only in Opisthokonta and three other protists[50]. In opisthokonts, the CatSper subunits enable the assembly of specialized $Ca^{2+}$ influx channels and are involved in the signaling for sperm maturation and motility[50]. In *Carpediemonas*, the tmcB family and CatSper subunits could similarly have a role in signaling and locomotion pathways required for a sexual cycle. As proteins in the cAMP pathway and $Ca^{2+}$ signaling cooperate to generate a variety of complex responses, the presence of these systems in *Carpediemonas* species but absence in all other sampled metamonads is intriguing and deserves further investigation. Even if these systems are not directly involved in a sexual cycle, the presence of Hap2 and Gex1 proteins is strong evidence that *C. membranifera* can reproduce sexually. Interestingly, based on the frequencies of single nucleotide polymorphisms, *C. membranifera* is predicted to be haploid (Supplementary Fig. 9). If this is correct, its sexual reproduction should include the formation of a zygote followed by a meiotic division to regain its haploid state[51].

**Acquisition of replication and repair proteins by lateral gene transfer.** The absence of many components of canonical DNA replication, repair, and segregation systems in *Carpediemonas* species led us to investigate whether they had been replaced by analogous systems acquired by lateral gene transfer (LGT) from viruses or prokaryotes. We detected four Geminivirus-like replication initiation protein sequences in the *C. membranifera* genome but not in *C. frisia*, and helitron-related helicase endonucleases in both *Carpediemonas* genomes. All these genes were embedded in high-coverage eukaryotic scaffolds, yet all of them lack introns and show no evidence of gene expression in the RNA-Seq data. As RNA was harvested from log-phase actively replicating cell cultures, their lack of expression suggests it is unlikely that these acquired proteins were coopted to function in the replication of the *Carpediemonas* genomes. Nevertheless, the presence of Geminivirus protein-coding genes is intriguing as

these viruses are known, in other organisms (e.g., plants, insects), to alter host transcriptional controls and reprogram the cell cycle to induce the host DNA replication machinery[52,53]. We also detected putative LGTs of Endonuclease IV, RarA, and RNAse H1 from prokaryotes into a *Carpediemonas* ancestor (Supplementary Information and Supplementary Figs. 10, 11, 12). Of these, RarA is ubiquitous in bacteria and eukaryotes and acts during replication and recombination in the context of collapsed replication forks[54]. Interestingly, *Carpediemonas* appears to have lost the eukaryotic ortholog and only retains the acquired prokaryotic-like RarA, a gene that is expressed (i.e., transcripts are present in the RNA-Seq data). RNAse Hs are involved in the cleavage of RNA from RNA:DNA hybrid structures that form during replication, transcription, and repair, and, while eukaryotes have a monomeric RNAse H1 and a heterotrimeric RNAse H2, prokaryotes have either one or both types. Eukaryotic RNAse H1 removes RNA primers during replication and R-loops during transcription and also participates in HR-mediated DSB repair[55]. The prokaryotic homologs have similar roles during replication and transcription[56]. *C. membranifera* lacks a typical eukaryotic RNAse H1 but has two copies of prokaryotic homologs. Both are located in scaffolds comprising intron-containing genes and have RNA-Seq coverage, clearly demonstrating that they are not from prokaryotic contaminants in the assembly.

## Discussion

The reductive evolution of the DNA replication, repair, and segregation systems and the low retention of proteins in the BUSCO dataset in metamonads demonstrate that substantial gene loss has occurred (Supplementary Information), providing additional evidence for streamlining of gene content prior to the last common ancestor of Metamonada[14–16]. However, the patchy distribution of genes within the group suggests an ongoing differential reduction in different metamonad groups. Such reduction—especially the absence of systems such as the ORC, Cdc6, and Ndc80 complexes in *Carpediemonas* species—demands an explanation. Whereas the loss of genes from varied metabolic pathways is well known in lineages with different lifestyles[57–59], loss of cell cycle, DNA damage sensing, and repair genes in eukaryotes is very rare. New evidence from yeasts of the genus *Hanseniaspora* suggests that the loss of proteins in these systems can lead to genome instability and long-term hypermutation leading to high rates of sequence substitution[57]. This could also apply to metamonads, especially fornicates, which are well known to have undergone rapid sequence evolution; these taxa form a highly divergent clade with very long branches in phylogenetic trees[19,60] (Supplementary Fig. 1). Most of the genes that were retained by Metamonada in the various pathways we examined were divergent in sequence relative to homologs in other eukaryotes and many of the gene losses correspond to proteins that are essential in model system eukaryotes. Gene essentiality appears to be relative and context-dependent, and some studies have shown that the loss of "indispensable" genes could be permitted by evolving divergent pathways that provide similar activities via chromosome stoichiometry changes and compensatory gene loss[57,58,61].

The patchy distribution of genes from different ancestral eukaryotic pathways suggests that the last common ancestor of Metamonada had a broad gene repertoire for maintaining varied metabolic functions under fluctuating environmental conditions offered by diverse oxygen-depleted habitats. Although the loss of proteins and genomic streamlining are well known in parasitic diplomonads[14,15], the Fornicata, as a whole, tend to have a reduced subset of the genes that are commonly found in core eukaryotic pathways. In general, such gene content reduction can

partially be explained as the result of historical and niche-specific adaptations[62]. Yet, given that (1) genome maintenance mostly depends on the cell cycle checkpoints, DNA repair pathways, and their interactions[13], (2) several missing proteins related to these pathways were present in the last common ancestor of metamonads, (3) aneuploidy and high overall rates of sequence evolution have been observed in metamonads[63,64], and (4) the loss of DNA repair genes can be associated with substantial gene loss and sequence instability that apparently boosts the rates of sequence evolution[57], it is likely that genome evolution in the Fornicata clade, in particular, has been heavily influenced by their error-prone DNA maintenance mechanisms. The DNA replication, repair, and segregation systems are more complete in non-fornicate metamonads suggesting that genome evolution in these organisms has been less affected as consequence.

Origin-independent replication has been observed in the context of DNA repair (reviewed in ref. [9]) and in origin-deficient or -depleted chromosomes in yeast[65]. These studies have highlighted the lack of (or reduction in) the recruitment of ORC and Cdc6 onto the DNA, but no study to date has documented regular eukaryotic DNA replication in the absence of genes encoding these proteins. While it is possible that highly divergent versions of ORC and Cdc6 are governing the recognition of origins of replication and replication licensing in *Carpediemonas* species, we have no evidence for this. Instead, our findings suggest the existence of an as-yet-undiscovered underlying eukaryotic system that can accomplish eukaryotic DNA replication initiation and licensing. The existence of such a system has in fact already been suspected given that: (1) Orc1- or Orc 2-depleted human cells and mouse-Orc1 and fruit-fly ORC mutants are viable and capable of undergoing replication and endoreplication[66–68] and (2) origin-independent replication at the chromosome level has been reported[65,69,70]. We propose a non-canonical DNA replication hypothesis in which *Carpediemonas* species utilize a replication system based on a Dmc1-dependent HR mechanism that is origin-independent, and mediated by RNA:DNA hybrids. Here, we first summarize evidence that such a mechanism is possible based on what is known in model systems and then present a model as to how it might occur in *Carpediemonas*.

During replication and transcription, the HR complexes, RNAse H1, and RNA-interacting proteins are recruited onto the DNA to assist in its repair[31]. Remarkably, experiments show that HR is able to carry out full genome replication in archaea, bacteria, viruses, and linear mtDNA[70–73], with replication fork progression rates that are comparable to those of regular replication[74]. A variety of *cis* and *trans* homologous sequences (e.g., chromatids, transcript-RNA, or -cDNA) can be used as templates[24,33], and their length as well as the presence of one or two homologous ends likely influence a recombination execution checkpoint that decides which HR sub-pathway is utilized[75]. For example, in the absence of a second homologous end, HR by Rad51-dependent break-induced replication (BIR) can either use a newly synthesized DNA strand or independently invade donor sequences, such that the initial strand invasion intermediate creates a migrating D-loop and DNA is synthesized conservatively[24,75]. Studies have found that BIR does not require the assembly of an ORC complex and Cdc6 but the recruitment of the Cdc7, loading of MCM helicase, firing factors and replicative polymerases are needed for assembling the pre-RC complex[24,75]. The requirement of MCM for BIR was questioned, as Pif1 helicase was found to be essential for long-range BIR[38]. However, recent evidence shows that MCM is typically recruited for unwinding DNA strands during HR[76] and is likely needed together with Pif1 to enhance processivity. All these proteins may also operate during origin-independent transcription-initiated replication (TIR), a still-enigmatic mechanism that

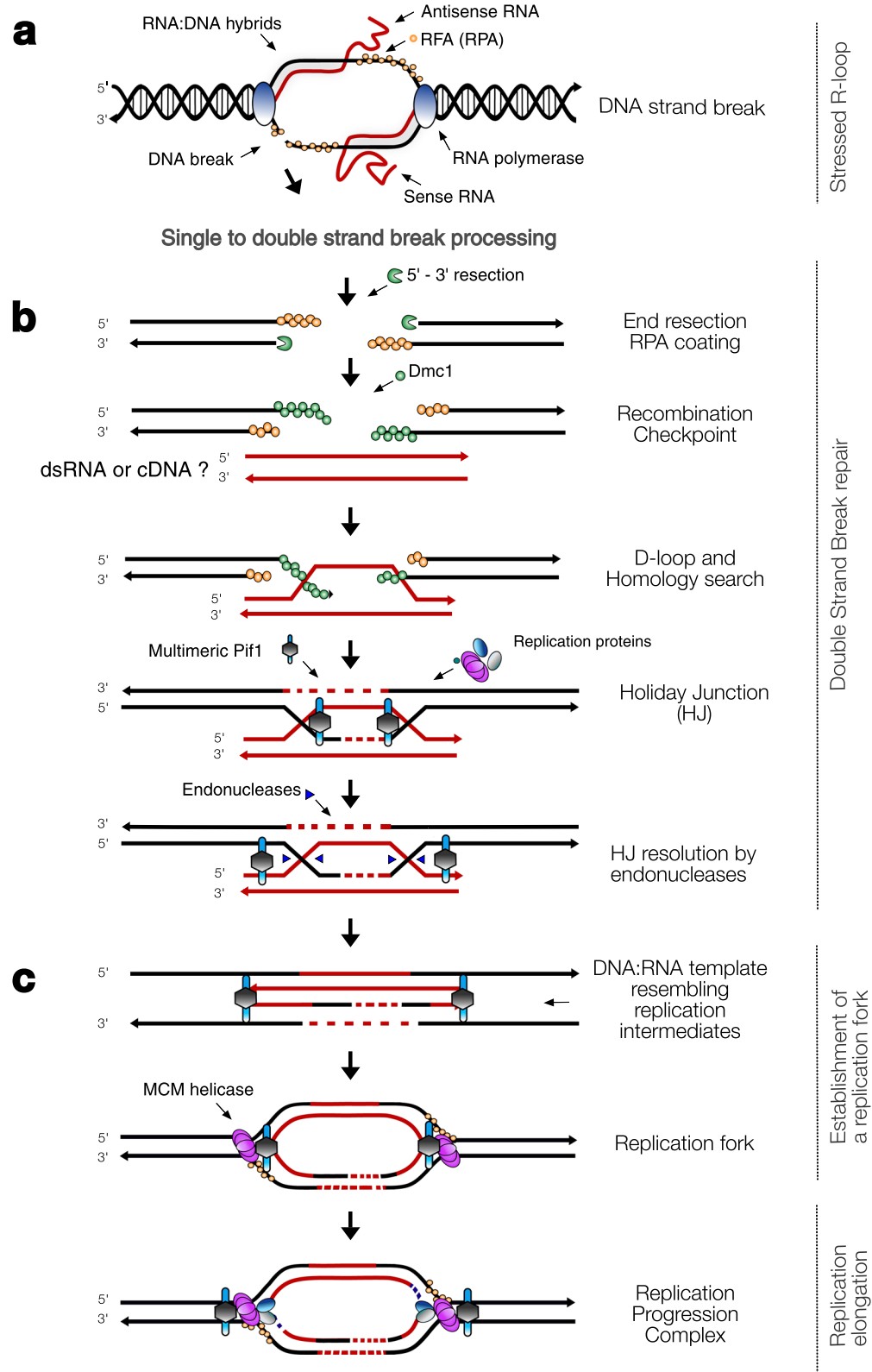

is triggered by DR-loops resulting from RNA:DNA and DNA:DNA hybrids during transcription[9,10,77].

Considering the complement of proteins in *Carpediemonas* species discussed above, and that RNA:DNA hybrids are capable of promoting origin-independent replication in model systems[10,32], we suggest that a Dmc1-dependent HR replication mechanism is enabled by an excess of RNA:DNA hybrids in these organisms. In such a system, DSBs generated in stressed transcription-dependent DR-loops[77] could be repaired by HR with either transcript-RNA- or transcript-cDNA-templates and the de novo assembly of the replisome as in BIR (Fig. 4). The establishment of a replication fork could be favored by the presence of *Carpediemonas*-specific Pif1-like homologs, as these raise the possibility of the assembly of a multimeric Pif1 helicase with

**Fig. 4 Hypothesis for Dmc1-dependent DNA replication in *Carpediemonas*. a** Full chromosome replication starts at multiple DR-loops undergoing sense and antisense transcription[77,93] in a highly transcribed locus that experiences DNA breaks, triggering DSB checkpoint control systems to assemble HR complexes and the replication proteins near the lesions[10,31,94–96]. **b** Once the damage is processed into a DSB, end resection creates an overhang, and the strands are coated with replication protein A (RPA), and the recombinase Dmc1. A recombination checkpoint decides the HR sub-pathway to be used[75], then strand invasion of a broken end is initiated into a transcript-RNA or -cDNA template[32,34]*; followed by the initiation and progression of DNA synthesis with the aid of Pif1 helicase. This leads to the establishment of a double Holliday Junction (HJ) which can be resolved by endonucleases (e.g., Mus81, Flap, and Mlh1/Mlh3). The lack of Chk1 may result in mis-segregation caused by aberrant processing of DNA replication intermediates by Mus81[40]. Given the shortness of the RNA or cDNA template, most possible HJ resolutions, except for the one depicted in the figure, would lead to the loss of chromosome fragments. The HJ resolution shown would allow steps shown in panel **c**. **c** A multimeric *Carpediemonas* Pif1-like helicase is bound to the repaired DNA as well as to the template. Here, the shortness of the template could resemble a replication intermediate that could prompt the assembly of a fully functional replication fork. Dark blue fragments on ends of the bottom figure represent Okazaki fragments. *Notes: Polymerases α and δ are able to incorporate the correct nucleotides using RNA template[33]; pol θ is able to reverse transcribe RNA[34]; RNAse H2 excise ribonucleosides and replaces them with the correct nucleotide.

increased capability to bind multiple sites on the DNA, thereby facilitating DNA replication processivity and regulation[37]. Note that the foregoing mechanisms will work even if *Carpediemonas* species are haploid as seems likely based on the SNP data. Since most elements of our proposed model are common to all eukaryotes, we speculate it has the potential to occur across eukaryotic diversity in addition to the canonical ORC-based system. The loss of Rad51A and the duplication of Dmc1 recombinases suggests that a Dmc1-dependent HR mechanism was likely enabled in the last common ancestor of Fornicata and this mechanism may have become the predominant replication pathway in the *Carpediemonas* lineage after its divergence from the other fornicates, ultimately leading to the loss of ORC and Cdc6 proteins.

DNA replication licensing and firing are temporally separated (i.e., they occur late M phase to G1/S transition, and S phases, respectively) and are the principal ways to counteract damaging over-replication[6]. As S-phase is particularly vulnerable to DNA errors and lesions, its checkpoints are likely more important for preventing genome instability than those of G1, G2, or SAC[78]. Dysregulation is anticipated if no ORC/Cdc6 are present as licensing would not take place and replication would be blocked[25]. Yet this clearly does not happen in *Carpediemonas*. This implies that during the late G1 phase, activation by loading the MCM helicase has to occur by an alternative mechanism that is still unknown but might already be in place in eukaryotes. Such a mechanism has long been suspected as it could explain the overabundance and distribution patterns of MCM on the DNA (i.e., the MCM paradox[79]).

In terms of the regulation of M-phase progression, the divergent nature of the KT in *C. membranifera* could suggest that it uses different mechanisms to execute mitosis and meiosis. It is known that in *Carpediemonas*-related fornicates such as retortamonads and in diplomonads, chromosome segregation proceeds inside a persisting nuclear envelope, with the aid of intranuclear microtubules, but with the mitotic spindle nucleated outside the nucleus (i.e., semi-open mitosis)[64]. Although mitosis in *Carpediemonas* has not been directly observed, these organisms may also possess a semi-open mitotic system such as the ones found in other fornicates. Yet how the *Carpediemonas* KT functions in the complete absence of the microtubule-binding Ndc80 complex remains a mystery; it is possible that, like in kinetoplastids[48], other molecular complexes have evolved in this lineage that fulfill the roles of Ndc80 and other KT complexes.

Interestingly, a potential repurposing of SAC proteins seems to have occurred in the diplomonad *G. intestinalis*, as it does not arrest under treatment with microtubule-destabilizing drugs and Mad2 localizes to a region of the intracytoplasmic axonemes of the caudal flagella[47]. Other diplomonads have a similar SAC protein complement that may have a similar non-canonical

function. In contrast to diplomonads, our investigations (Fig. 3) suggest that *Carpediemonas* species could elicit a functional SAC response, although microtubule-disrupting experiments during mitosis will be needed to prove its existence.

In addition to the aforementioned apparent dysregulation of checkpoint controls in *Carpediemonas* species, alternative mechanisms for chromosome condensation, spindle attachment, sister chromatid cohesion, cytokinesis, heterochromatin formation, and silencing and transcriptional regulation could also be expected in this organism due to the absence of ORC and Cdc6 (reviewed in refs. [27,80]). All of the absences of canonical eukaryotic systems we have described for *Carpediemonas* suggest that a very different cell cycle has evolved in this free-living protistan lineage. This underscores the fact that our concepts of universality and essentiality rely on studies of a very small subset of organisms. Since the actual DNA replication mechanism in *Carpediemonas* species remains undiscovered, the development of *C. membranifera* as a model system has great potential to enhance our understanding of fundamental DNA replication, repair, and cell cycle processes. For instance, our replication hypothesis could, in principle, be studied by targeted knockouts (or "knockdowns") of one, or both, of the *DMC1* genes. The expectation would be that the single knockout would show lower fitness than the wild type, whereas the double knockout strain would not be viable unless rescued by a plasmid-encoded tagged Dmc1 protein, or genomically-inserted gene whose expression could be controlled. Such experiments could be complemented with deep genome sequencing to obtain and compare replication profiles at a log and stationary phases (i.e., estimation of the ratio of uniquely mapped reads in each phase)[70,81], as well as differential gene expression experiments to determine whether the replication profiles are correlated with highly transcribed loci indicating origin-independent replication initiation. Once tools for genetic manipulation and cell biology are developed for *Carpediemonas*, experimental studies, including those described above, can be conducted to test the replication hypothesis advanced here (Fig. 4). This will also help us to determine if the unusual systems underpinning *Carpediemonas* DNA replication, segregation, and cell cycle are unique to this organism, are potentially present in other metamonads, or represent a more general alternative replication mechanism found across eukaryotic diversity.

## Methods

**Sequencing, assembly, and protein prediction for *C. membranifera*.** DNA and RNA were isolated from cultures of *C. membranifera* BICM strain (see details in Supplementary Information). Sequencing employed Illumina short paired-end and long read (Oxford Nanopore MinION) technologies. For Illumina, extracted, purified DNA and RNA (i.e., cDNA) were sequenced on the Hiseq 2000 (150 × 2 paired-end) at the Genome Québec facility. Illumina reads were quality trimmed (Q = 30) and filtered for length (>40 bp) with Trimmomatic v0.39[82]. For MinION, the library was prepared using the 1D native barcoding genomic DNA (SQK-

LSK108 with EXP-NBD103) protocol (NBE_9006_v103_revP_21Dec2016). The final library (1070 ng) was loaded on an R9.4 flow cell and sequenced for 48 h on the MinION Mk1B nanopore sequencer. Long read processing, genome assembly, and decontamination methodologies are reported in Supplementary Information.

RNA-Seq reads were used for genome-independent assessments of the presence of the proteins of interest and to generate intron hints for gene prediction. For the independent assessments, we obtained both a de novo and a genome-guided transcriptome assembly with Trinity v2.5.0[83]. Open reading frames were translated with TransDecoder v5.5.0 (www.github.com/TransDecoder) and were included in all of our analyses. Gene predictions were carried out as follows: repeat libraries were obtained and masked with RepeatModeler v1.0 and RepeatMasker v4.0.7 (http://www.repeatmasker.org). Then, RNA-Seq reads were mapped onto the assembly using Hisat2 v2.1.0[84], generating a bam file for GenMarkET 4.38[85]. This resulted in a list of intron hints used to train Augustus v3.2.3[86]. The genome-guided assembled transcriptome, genomic scaffolds, and the newly predicted proteome were fed into the PASA v2.3.3 pipeline[87] to yield a more accurate set of predicted proteins. Finally, the predicted proteome was manually curated for the proteins of interest.

**Genome size, completeness, ploidy assessments, and phylogenetic placement**. We estimated the completeness of the draft genome by (1) using the k-mer based and reference-free method Merqury v1.3[20], (2) calculating the percentage of transcripts that aligned to the genome, and (3) employing the BUSCO v3.0.2[88] framework. For method 1, all paired-end reads were used to estimate the best k-mer and create "meryl" databases necessary to apply Merqury[20]. For method 2, transcripts were mapped onto the genome using BLASTn v.2.7.1 and exonerate v2.54.1[89]. For method 3, the completeness of the draft genome was evaluated in a comparative setting by including the metamonads and using the universal single-copy orthologs (BUSCO) from the Eukaryota (odb9) and protist databases (https://busco.ezlab.org/), which contain 303 and 215 proteins, respectively. Each search was run separately on the assembly and the predicted proteome for all these taxa. Unfortunately, both BUSCO database searches yielded false negatives in that several conserved proteins publicly reported for *T. vaginalis*, *G. intestinalis*, and *Spironucleus salmonicida* were not detected due to the high divergence of metamonad homologs. Therefore, genome completeness was reassessed with a phylogeny-guided search (Supplementary Information).

The ploidy of *C. membranifera* was inferred by (i) counting k-mers with Merqury[20] and (ii) mapping 613,266,290 Illumina short reads to the assembly with Bowtie v2.3.1[90] and then using ploidyNGS v3.0[91] to calculate the distribution of allele frequencies across the genome. A site was deemed to be heterozygous if at least two different bases were present and there were at least two reads with the different bases. Positions with less than 10× coverage were ignored. For completion, we also assessed the phylogenetic placement of *C. membranifera* and *C. frisia* within Metamonada as described in Supplementary Information.

**Functional annotation of the predicted proteins**. Our analyses included the genomes and predicted proteomes of *C. membranifera* (reported here) as well as publicly available data for nine additional metamonads and eight other eukaryotes representing diverse groups across the eukaryotic tree of life (Fig. 1, Table 1, and Supplementary Information). Orthologs from each of these 18 predicted proteomes were retrieved for the assessment of core cellular pathways, such as DNA replication and repair, mitosis and meiosis, and cell cycle checkpoints. For *C. membranifera*, we included the predicted proteomes derived from the assembly plus the six-frame translated transcriptomes. Positive hits were manually curated in the *C. membranifera* draft genome. A total of 367 protein queries were selected based on an extensive literature review and prioritizing queries from taxa in which they had been experimentally characterized. The identification of orthologs was as described for the BUSCO proteins but using these 367 queries for the initial BLASTp v.2.7.1 (Supplementary Information), except for KT, SAC, and anaphase promoting complex-related genes (APC/C). For these, previously published HMMs with cut-offs specific to each orthologous group (see ref. [58]) were used to query the proteomes with HMMER v3.1b2[29]. A multiple sequence alignment that included the newly-found hits was subsequently constructed with MAFFT v7.310[92] and was used in HMM searches for more divergent homologs. This process was iterated until no new significant hits could be found. As we were unable to retrieve orthologs of a number of essential proteins in the *C. membranifera* and *C. frisia* genomes, we embarked on additional more sensitive strategies to detect them using multiple different HMMs based on aligned homologs from archaea, metamonads, and broad samplings of taxa. Individual PFAM v33.1 domains were searched for in the genomes, proteome, and translated transcriptomes with e-value thresholds of $10^{-3}$ (Supplementary Information). To rule out that failure to detect these proteins was due to insufficient sensitivity of our methods when applied them to highly divergent taxa, we queried 22 extra eukaryotic genomes with demonstrated high rates of sequence evolution, genome streamlining, or unusual genomic features (Supplementary Data 3, Supplementary Fig. 4, and Supplementary Information). Possible non-predicted or mispredicted genes were investigated using tBLASTn searches of the genomic scaffolds, unassembled reads, and six-frame translation searches with HMMER. Also, as DNA replication and repair genes could have been acquired by LGT into *Carpediemonas* species from prokaryotes or viruses, proteins from the DNA replication and repair categories whose best matches were to

prokaryotic and viral homologs were subjected to phylogenetic analysis using the methods described for the phylogeny-guided BUSCO analysis and using substitution models specified in the legend of each tree (Supplementary Information).

**Reporting Summary**. Further information on research design is available in the Nature Research Reporting Summary linked to this article.

## Data availability
The genome assembly generated in this study has been deposited in GenBank under BioProject PRJNA719540 and WGS accession number JAHDYR000000000. RNA-seq reads have been deposited in NCBI Sequence Read Archive with accession number SRR15678499. High-resolution versions of the figures embedded in the Supplementary Information are available at Dryad (https://doi.org/10.5061/dryad.wh70rxwnv).

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

## Acknowledgements

The majority of this work was supported by a Foundation grant FRN-142349, awarded to A.J.R. by the Canadian Institutes of Health Research. Archibald Lab contributions to this study were supported by a Discovery Grant from the Natural Sciences and Engineering Research Council of Canada (RGPIN 05871-2014). E.C.T. acknowledges support from a Herchel Smith Postdoctoral Fellowship (University of Cambridge, UK), and the Dutch Science Organisation (VI.Veni.202.223). We would like to thank Ryan Wick for his helpful comments on genome assembly error correction.

## Author contributions

D.E.S.-L. and A.J.R. conceived the study. J.J.-H. and M.K. grew cultures, extracted nucleic acids, and carried out in-house sequencing. D.E.S.-L., B.A.C., E.C.T., Z.Y., J.S.S.-L., L.G.-L., S.K.W., G.J.P.L.K., J.M.A., A.G.B.S., and A.J.R. analyzed and manually curated the genomic data. E.C.T. and D.E.S.-L. made the figures. D.E.S.-L. and A.J.R. led the writing of the manuscript with input from all authors. All documents were edited and approved by all authors.

## Competing interests

The authors declare no competing interests.
