## [Peer Review File · Nature Communications]

Genomic analysis finds no evidence of canonical eukaryotic DNA processing complexes in a free-living protistAnswer to Reviewer's comments (initial submission)******* Reviewer #1 *******

Reviewer #1 (Remarks to the Author):

The paper entitled "A free-living protist that lacks canonical eukaryotic DNA replication and segregation systems" is a valuable, fascinating and well-written study of the conserved eukaryotic cellular systems across the tree of life, including some of the neglected lineages of eukaryotes such as Metamonada. The authors carefully analysed the draft genome of *Carpediemonas*, a free-living representative of Metamonada, to better understand unprecedented losses of DNA replication and segregation systems components in this lineage. Surprisingly, *Carpediemonas* turned out to be even more reduced than expected.

Most of the sequenced genomes of Metamonada belong to parasites, such as *Giardia* or *Trichomonas*. Therefore a free-living representative is a valuable addition for this and further studies on the whole group. Surprisingly in some respects, genomes and cellular systems of non-parasitic Metamonada lineages are more streamlined than parasitic ones. For instance, *Monocercomonoides exilis* is a metamonad with no mitochondrion, and in this paper, the authors have convincingly shown that *Carpediemonas* lost large complexes of the canonical eukaryotic DNA replication and segregation systems.

Papers on model organisms dominate eukaryotic cell biology studies. However, to better understand the eukaryotic cell evolution and functioning, the non-model organisms representing other lineages than plants, animals and fungi are critical. As have been previously shown, Metamonada might represent exciting deviations from the canonical eukaryotic cell systems, and they might become models to study cell biology. *Carpediemonas* is one of those fascinating cases highlighting eukaryotic cell versatility beyond the model organisms. The presented study is of interest to evolutionary biologists and cell biologists due to its broad implications for understanding the eukaryotic cell systems evolution.

Reviewer #1 Comment 1.

This study is based on meticulous genome annotation and comparative analyses. I am fully convinced that the quality of the data and performed analyses undoubtedly prove the loss of large conserved complexes involved in the cell cycle. That is a fascinating discovery, but the narration is somehow disappointing. **a)** Throughout the text, there are suggestions that another system evolved in this lineage to cope with DNA replication/repair, but no alternative was proposed. Of course, if no alternative system was discovered so far, there is no definitive answer. **b)** Still, I would expect more detailed comments on the absence and presence of the particular components since most of them are also absent in one or more other lineages suggesting that those are not critical for the cell functioning. **c)** I appreciate the discussion on what might be an alternative system, how it could be possibly discovered, and what are the chances that other organisms use some alternative systems too. It should be more clearly stated in the abstract and conclusions that such a system still awaiting discovery.

Response to Reviewer #1's comment 1:

a) Full DNA replication in eukaryotes has only been observed to be origin-dependent and to use proteins and mechanisms of the replisome. Whereas there is no knowledge of a novel or alternative system for full DNA replication in eukaryotes, recent experimental evidence indicates that replication could start in an origin-independent manner as response to DNA damage and could lead to either partial or full chromosome replication. Such mechanisms, mainly break induced replication (BIR) and transcription-initiated replication (TIR), were mentioned in our initial submission, and served as the basis for our

hypothesis on how *Carpediemonas* species could replicate its DNA (see Initial Submission (IS) lines 325-373 and Fig 4). Since most elements of our proposed model are common to all eukaryotes, we think that our hypothesised mechanism could potentially occur more broadly across eukaryotic diversity, as a supplementary system to the well-known ORC-based system. To improve clarity, we revised the Discussion to state: "We propose a non-canonical DNA replication hypothesis in which *Carpediemonas* species utilize a replication system based on a Dmc1-dependent HR mechanism that is origin-independent and mediated by RNA:DNA hybrids." (Revised Submission (RS) lines 362-365), which continues with two paragraphs showing the current experimental evidence as well as a detailed explanation of the hypothesis.

- b)** We agree with the reviewer that the patchy presence of some proteins in several taxa suggest that these may not be essential, and that their function may be carried out by other proteins. However, it is also likely that there is some impairment of some functions in these organisms relative to those taxa that possess these systems. Due to space constraints, we have included a small amount of new information in RS lines 138-140 in the main text and included a new section in the Supplementary Information with a considerably expanded discussion (lines 165-217).
- c)** As mentioned in **a)** above, we did propose a potential alternative mechanism as a hypothesis in the IS. Nevertheless, in the RS, we have re-worked the corresponding paragraphs to make this clearer. As the reviewer suggests, we have included a paragraph commenting that such a system still awaits discovery and how our proposed hypothesis could, in principle, be tested (RS lines 443-454). Also, a brief statement on the possibility that other organisms may use this alternative system has been included (RS lines 397-399). Also, as the reviewer suggests, we have clarified in the abstract that the replication mechanism is still undiscovered (RS line 45-46).

Reviewer #1 Comment 2

I would also expect that since Metamonada is so prone to losses of crucial cell systems, a discussion of the evolution of the whole group would be in place. Perhaps the entire group is somehow compromised by the unstable DNA replication/repair systems?

Response to Reviewer #1's comment 2:

We had a paragraph dedicated to this topic (focusing on mostly the Fornicata) in the IS Discussion section lines 311-323, now in RS lines 330-342). To this we have added "The DNA replication, repair, and segregation systems are more complete in non-fornicate metamonads suggesting that genome evolution in these organisms has been less affected as a consequence." (RS lines 348-350).

Reviewer # 1 Specific comment 1

What is the phylogenetic position of *Carpediemonas*? Please, provide a more detailed phylogenetic analysis of Metamonada or at least the description of previous phylogenies. The topology might have implications for the evolution of features. Therefore, some discussion on the phylogeny should be provided.

Response to Reviewer #1's specific comment 1:

The phylogenetic position of *Carpediemonas* has been previously determined in phylogenomic and phylogenetic analyses in Leger et al. (2017) and Hamann et al (2017), respectively and those analyses were the basis for our original schematic trees in Fig. 1 and 3. To address the reviewer's request, in the RS, we have now added a phylogenomic analyses to our study. The resulting tree of Metamonada is shown in RS Supplementary Fig. 1 and briefly discussed in the Results section (RS lines 121-129, RS Suppl. Fig 1). Our

new phylogeny recovers the same relationships as previously published studies, although the taxon sampling is improved. As a result, none of our original conclusions about the timing of the gene losses are affected.

Hamann, E. *et al* (2017). <https://doi.org/10.1038/ismej.2016.197>

Leger, M. *et al* (2017) <https://www.ncbi.nlm.nih.gov/pubmed/28474007>

Reviewer # 1 Specific comment 2

Authors mentioned that other metamonads, such as *Giardia* and *Trichomonas*, are highly divergent, and that might be caused by less effective DNA replication and/or repair systems in those lineages. What about *Carpediemonas*? Is it also a long-branching metamonad?

Response to Reviewer #1's specific comment 2:

Yes, *Carpediemonas* are long-branching metamonads, although not quite as divergent as the parasitic diplomonads like *Giardia* and *Spironucleus* or the secondarily free-living species *Trepomonas* sp. (RS lines 124-129, new Suppl. Fig. 1). The statement about DNA replication/repair impact in RS lines 339-348 (not highlighted) covers *Carpediemonas* species.

Reviewer # 1 Specific comment 3

Figure 1 - Since that is an essential figure, it should be more readable. Of course, I can enlarge it to read the numbers, but then the whole tree is not visible. I am a bit confused by the symbols on the branches. Why are some circles in colour? Are those corresponding with particular complexes? That is not very well visible since the circles are pretty small. I am also confused by the question mark in the orange circle at the Metamonada and Discoba branch. What does it mean?

Response to Reviewer #1's specific comment 3:

In the IS the figures were embedded in the body of the manuscript at lower resolution, which made their readability difficult. In the RS, we have increased font sizes and improved various visual aspects to enhance legibility and have also provided the figures separately in high-resolution. In addition, we clarified the meaning of the circles and question marks in the legend (Lines 788-809).

Reviewer # 1 Specific comment 4

The name *Fornicata* should be explained in the text.

Response to Reviewer #1's specific comment 4:

The name *Fornicata* is now mentioned and defined in the Results section that discusses the phylogenomic analysis (RS lines 124-127).

Reviewer # 1 Specific comment 5

Most of the proteins have a patchy distribution, which was not commented on in details. Especially the Metamonada lineage requires more information on how the analysed systems work in other species than *Carpediemonas*.

Response to Reviewer #1's specific comment 5:

Given the space constraints and the number of molecular systems we have investigated it is not possible to discuss in detail each of the molecular systems in each metamonad lineage. Instead, we have prioritized discussing the *Carpediemonas* replication/segregation systems as they are the most distinctive. However, as mentioned above, we have added a substantial discussion about the most relevant findings regarding the other metamonads to the Supplementary Information (lines 165-217)

Reviewer # 1 Specific comment 6

The presence/absence pattern of ORC/cdc6 proteins is slightly different than in the previous publications on *Trichomonas*, *Giardia* or *Monocercomonoides* genomes – are those newly discovered proteins? If yes, perhaps that should be commented. For example, Orc5 was not previously reported in *Trichomonas* and *Monocercomonoides*, and on the contrary, Orc1 was reported in *Giardia*. Please comment on your newly obtained results and the possible source of differences with previous results.

Response to Reviewer #1's specific comment 6:

In the case of the Orc5 proteins, we report them here for first time. Previous studies have applied state-of-the-art search strategies that include direct and reciprocal blast and Hidden Markov Models (HMM). We used these same techniques, but the major reason for the successful retrieval of proteins in our work is our use of HMMs from taxa-enriched PANTHERs. These HMMs encompass broad amino acid sequence diversity that improves the sensitivity of HMMER-searches in detecting distant homologs. As described in the methods section, once a protein was retrieved with these HMMs, we validated its orthology by phylogenetic and protein domain architecture reconstruction. In the case of *Giardia*, the most-up-to-date assignment of the 'Orc1' proteins is Orc1/Cdc6 (GL50803_17103, GL50581_2939, GMRT_20170, GMRT_24689) or hypothetical proteins (GL50803_8922, GL50581_419) (<https://giardiadb.org/>). We were able to classify the hypothetical proteins as members of the Orc1/Cdc6 protein family. Our phylogenetic analysis demonstrates that it is difficult to establish which proteins are Orc1 and which are Cdc6; hence, we referred to them as Orc1/Cdc6-like (RS lines 149-150) and RS-supplementary Fig. 3 panel D). Since our annotation continues to reflect the ambiguity in the classification of these proteins, we previously avoided commenting on it. However, to remedy this, we have added two sentences commenting on the *de novo* retrieval of the above-mentioned proteins and included a new figure containing a phylogenetic tree for Orc5 proteins (RS lines 143-148, Supplementary Fig. 2).

We thank the Reviewer#1 for all their comments, and we believe these suggestions have helped us to improve our manuscript significantly.

******* Reviewer #2 *******

- What are the noteworthy results?

This manuscript presented a high quality *Carpediemonas* genome, and with thorough analysis, it shows that this free-living protist lacks canonical eukaryotic DNA replication and segregation system.

- Will the work be of significance to the field and related fields? How does it compare to the established literature? If the work is not original, please provide relevant references.

Yes, this work is of significant value to the protist field being the first near-complete free-living metamonad genome. Interestingly, this genome reveals that this free-living protist is further streamlined in many perspectives, like it lacks many key components in DNA replication and segregation systems, and this is the first eukaryote found so far that have lost this large suite of conserved complexes, suggesting it reply on novel or alternative set of mechanisms to carry out these fundamental processes. I find this work original and of high quality.

- Does the work support the conclusions and claims, or is additional evidence needed?

The authors obtained a draft genome, and couldn't find those essential genes involved in the essential cellular process. Therefore, the authors did extensive data mining in the genome assembly, unassembled long reads.

To prove something is not there is harder than to prove something is there. But I think the authors did thorough investigation to prove the lost genes were truly lost.

- Are there any flaws in the data analysis, interpretation and conclusions? - Do these prohibit publication or require revision?

I found the data analysis in the manuscript of good quality, the interpretation were based on the analysis, and the conclusions were arrived logically.

Reviewer #2 Comment 1

1. It's sound strategy to use high quality Illumina reads to correct error prone MinION assembly, but I'm curious why the choice of Unicycler, which is a hybrid assembly pipeline for bacterial genome. I also wonder how much errors were corrected and what kinds of errors were corrected?

Response to Reviewer #2's comment 1:

The Unicycler pipeline is promoted as being specific for bacteria because its assembly process deals with the circular organization of prokaryotic chromosomes. However, the pipeline contains different stand-alone tools that are independent of the assembly process. As noted in the methods, we only conducted error correction with Unicycler and did not use it for assembly. We used the stand-alone tool called 'unicycler_polish' which uses short reads to do iterative correction on a provided assembly, functioning as a 'wrapper' for the Pilon and bowtie2 programs. Pilon is usually used in combination with bowtie2 for polishing/error-correction to correct sequences from long read technologies; however, our choice was to use them as implemented in 'unicycler_polish' because **1)** *C. membranifera's* genome is small (the code is not optimized for large assemblies) and predicted to be haploid (a personal communication with R. Wick, the program developer, suggested that its use on a haploid eukaryotic genome should work fine), and **2)** the stand-alone tool uses an iterative approach to evaluate whether or not there were improvements in each round of corrections; hence, it determines if additional correction rounds are needed. We would like to mention that we did not use the ALE-guided correction of larger variants output (one of the outputs) as the final error-corrected assembly, instead, we used the output produced in round 8 (see corrections per round below). We realize that the original description in our Initial Submission (IS) requires the additional information we have mentioned here. Therefore, in addition to relocating the sentence in Supplementary information (re-allocation was due to limit on citations in the main text), we have addressed this comment in the Supplementary Information lines 66-77 of the Revised Submission (RS).

Corrections done by each round (broken down by the type of corrections) Note: this table is also presented as Supplementary Table F in the current revision

Round	Variants applied after the round	Homopolymer corrections	Insertions	Deletions	Substitutions
1	16804	12098	924	872	2910
2	543	143	60	74	266
3	191	44	29	20	98
4	101	19	11	6	65
5	85	12	4	5	64
6	55	6	4	6	39
7	47	6	2	2	37
8	10	4	0	2	4

Reviewer #2 Comment 2

2. What was the criteria for removal of prokaryotic contigs? Were they all from its food *Shewanella frigidimarinx*?

Response to Reviewer #2's comment 2:

Carpediemonas membranifera was grown with *Shewanella frigidimarina*, *Shewanella* sp. or *Vibrio* sp. isolate JH43 as food. Several *Vibrio* sp. and *Shewanella* sp assemblies, including that of *Shewanella frigidimarina*, are represented in the nt database that was used for decontamination. Most contigs in the assembly were very large, and contaminating sequences matched different prokaryotic taxa. The large contig size combined with high percentage identities and percentage coverages against nt facilitated the decontamination process. A contig was deemed as contaminant if $\geq 60\%$ of it hit only prokaryotic sequences with percentage identity $\geq 40\%$ and an e-value threshold of 10^{-3} . We have now added this information in the Supplementary Information (lines 77-79)

Reviewer #2 Comment 3

3. One thing that wasn't clear to me is that did you mine in the transcriptome data for the missing genes? If not, what's the reasoning.

Response to Reviewer #2's comment 3:

Yes, we did mine the transcriptome. This was mentioned in the methods section of the IS lines 434-437, now also in RS lines 473-476

Reviewer #2 Comment 4

- Is there enough detail provided in the methods for the work to be reproduced?

I found the methods detail described. However, software version/build information is sometimes missing. Also why are there links to some software packages but not others? I think consistency would be nice.

Response to Reviewer #2's comment 4:

We have added the version information and now all the packages are cited according to their reference paper (and additional links were eliminated) where this was possible. The exceptions were Albacore v2.3.3 (it is only available to ONT customers via their community site), plus TransDecoder v5.5.0, RepeatModeler v1.0, and RepeatMasker v4.0.7 as these have no associated papers.

Versions added to: Unicycler v0.4.4, Blast v2.7.1, samtools v1.11, RepeatMasker v4.0.7, Hisat2 v2.1.0, GenMarkET v4.38, PASA v2.3.3, Merqury v1.3, BUSCO v3.0.2, exonerate v2.54.1, ploidyNGS v3.0.

Reviewer #2's specific comment 1:

1. I couldn't retrieve the data using the provided private url.

Response to Reviewer #2's specific comment 1:

Perhaps the link got broken when it was added to the manuscript. We provide a shorter link here and in the manuscript (<https://tinyurl.com/Carpediemonas>). If this still does not work here is a backup link (<https://tinyurl.com/Carpbackup>). In any case, we submitted the data to NCBI and are waiting for the accession numbers, which will be immediately available to the scientific community.

Reviewer #2's specific comment 2:

2. The first 5 rows in Table 1 as well as the detailed 6 rows of repeat information were not at all mentioned in the main text. My feeling is that the manuscript is missing a paragraph to describe the genome in general, which would also be interesting for researchers interested in this genome. Or are you planning to do this genome summary in some other ways?

Response to Reviewer #2's specific comment 2:

We have a second paper that is in the final draft stage. In that paper we conduct a comprehensive comparative genomics analysis of *C. membranifera*'s genome and carried out a detailed description of the sort mentioned by the reviewer. Given the importance of our findings regarding the replication/segregation apparatus of *Carpediemonas* and the space constraints of the current manuscript, we have chosen not to elaborate further on general genome features, although we have briefly added some information in RS lines 100-102. We also have eliminated the information about repeats from the Table 1 because it is not relevant for the aim of this paper but will be reported in the second one.

Reviewer #2's specific comment 3:

I also think the authors should compare stats with the newest genomes in Table 1, for example there is a new better *Giardia intestinalis* A 50803 genome which was published around the same time as the *Giardia muris* genome. And for clarity and re-producing purpose, there should be references to the genomes used in the table.

Response to Reviewer #2's specific comment 3:

We obtained the proteome and draft genome of *G. muris* from co-author J. Jerlström-Hultqvist long before its publication; hence it was possible for us to include it in the current comprehensive analyses. Despite the similar publication timing of the new *G. muris* and *G. intestinalis* newest drafts, we did not have similar early access to the latter one. For that reason, the statistics provided in the Table 1 correspond to the actual *G. intestinalis* A 50803 genome version used in this study, and we consider that keeping these is more appropriate. We have added the references to all genome assemblies used in the corresponding section in the Supplementary information because there is a constraint in the number of references allowed in the main text. We also included a statement regarding the newest version of the *G. intestinalis* A 50803 genome (see Supplementary Information lines 106-108).

Reviewer #2's specific comment 4:

3. If shortening of text is desired, I found the last part of introduction (from 'Our analyses of genome' to the end) is basically a long summary of the results, hence could be removed or integrated into conclusion.

Response to Reviewer #2's specific comment 4:

We cannot remove the paragraph in question because it is a 'must have' according to the format of Nature Communications.

Reviewer #2's specific comment 5:

Thanks for the contribution of this high-quality genome to the community!

Response to Reviewer #2's specific comment 5:

We deeply appreciate all the comments and questions raised by the reviewer as they helped improve our manuscript. We hope *C. membranifera*'s genome assembly becomes a helpful resource for other researchers.

******* Reviewer #3 *******

Reviewer #3 (Remarks to the Author):

Salas-Leiva and colleagues report a detailed, high-quality comparative genomics analysis for the evolution of eukaryotic replication, repair and segregation proteins in metamonads. The most striking result they presented here is that *Carpediemonas* lack ORC, Cdc6, and other crucial replication components in eukaryotes which implies that there are alternative paths/methods to fulfill the jobs of these proteins.

Reviewer #3 Comment 1

Their result on *Carpediemonas* is highly interesting as it lacks ORC and Cdc6. The genomics analysis and bioinformatic strategies are convincing itself. However, it would be a better picture if there were experiments to support this analysis. For instance, there are algorithm predictions to localize proteins (in this case, the authors may not even need that given that we have a really good idea where DNA replication occurs in cells) which is followed by immunofluorescence or confocal imaging of the protein in the predicted area of the cell to confirm the algorithm predictions. Using another metamonad that shows the presence of these proteins (fully or partially) in their genomic analysis as a positive control, lack of ORC and Cdc6 can be demonstrated with an imaging experiment described above. More importantly, what replaces them, if any? Are there any candidates? If so, any approaches to validate their possible origin recognition/replication initiation function? These are all important questions waiting an answer.

Response to Reviewer #3's comment 1:

We are glad that the Reviewer finds our manuscript highly interesting and backed up with sound bioinformatic strategies; our work is the result of more than four years of intense and meticulous analyses. We agree that our findings have raised important questions that could potentially be tackled by follow-up experiments in *Carpediemonas*. Unfortunately, however, there are a series of formidable methodological challenges with this non-model organism that would need to be surmounted before such experimental approaches will be possible. In addition, some the specific questions posed run into the problem of

'providing a negative.' To make the magnitude of the difficulties clear, we 'unpack' each of these challenges below as they pertain to the reviewer's specific comments 1-3 (as labelled here; see below).

The experiment indicated by the Reviewer#3 in comment 1 is unlikely to shed light on the main questions we are addressing — namely the absence of canonical replication and segregation systems in *Carpediemonas*. Because the suggestion is to use immunofluorescence imaging to show that the ORC-like proteins in other metamonads function in the nucleus (as a positive control) and then conduct a similar imaging experiment in *Carpediemonas* to demonstrate the "lack of ORC and Cdc6". But to do this experiment, we would have to raise an antibody to the ORC-like protein in another metamonad and then use this as a heterologous antibody against *Carpediemonas* in an immunofluorescence labelling experiment. Unfortunately, heterologous antibodies are notorious for failing to bind to homologous proteins in distantly related organisms. Most currently known metamonads are highly divergent and distantly related to *Carpediemonas* species, as evidenced by their long branches in phylogenetic reconstructions (see Leger et al 2017 and our Supplementary Figure 1). So even if *Carpediemonas* species did have a similar candidate protein (which, we show, it does not), a lack of 'signal' would be expected because of failure of the antibody to recognize a distantly related antigen. The only information this would provide is a possible confirmation that these proteins function in the nuclei of other metamonads, which would be only a confirmatory result for these other metamonads but of little significance to this manuscript as it fails to address our main findings.

Leger et al (2017) <https://www.ncbi.nlm.nih.gov/pubmed/28474007>

Reviewer #3 Comment 2

Same applies for Ndc80 complex being absent. Given that Ndc80 is involved in a different molecular mechanism, different experimental approaches can be used to show the absence of it such as co-IP experiments (co-IP might be another way to be used to show the lack of ORC and Cdc6). These experiments would also help to identify other proteins that are (perhaps) unexpectedly involved in the process to replace Ndc80 function. I agree with the authors that it is quite a mystery that how these organisms have functioning kinetochores without Ndc80 and that's why it would be incredibly interesting to find out if there is at least a replacement protein (or a complex that overtakes Ndc80's function) yet alone a different mechanism.

Response to Reviewer #3's comment 2:

We are a bit confused by this comment. Since we cannot raise an antibody against proteins that are absent in *Carpediemonas*, we suspect the Reviewer is suggesting that we develop an antibody against one of these proteins in another metamonad instead, and then use it in *Carpediemonas* to do co-IP experiments to identify binding partners. Given the high divergence among metamonads mentioned above, this approach is expected to fail *a priori* for the same reasons we provide in our response above. Thus, a negative result would mean virtually nothing. In principle, it might be possible to develop antibodies against other potential kinetochore-related proteins in *Carpediemonas* and attempt co-IP but not for replication fork proteins. In the case of kinetochore-related proteins, we estimate that such an experiment would require at least 8 months of dedicated work, the development of multiple antibodies against multiple proteins (since, in our experience, there is approximately 30% success rate that antibodies work well in *both* westerns blots and immunofluorescence experiments; one needs both to confirm that the expressed protein is the one being recognized in the immunofluorescence microscopy work). Furthermore, it would

take more than 5 months to express the proteins and raise the antibodies and purify them, and then potentially months to optimize immunofluorescence microscopy fixation protocols to work with *Carpediemonas*, an anaerobic protist that grows in seawater in mixed bacterial culture. The Roger lab has extensive experience with developing the latter protocols and conducting these experiments for other 'novel' anaerobic marine protists, and they involve time-consuming (*i.e.*, multi-year) trial and error approaches that frequently fail. In the case of replication fork proteins, such an approach would not work because the replication proteins we found are not specific for initiating DNA replication, since they are also involved in DNA repair by homologous recombination (HR) (Lydeard et al. 2010). Hence, it would not be possible to differentiate between origin-independent replication started by HR or house-keeping HR.

Lydeard et al. (2010) <https://doi.org/10.1101/gad.1922610>

Reviewer #3 Comment 3

Another interesting part of the study is SAC signaling/response. The authors report that *Carpediemonas* species could elicit a functional SAC response, but it might not be kinetochore related. As the authors also suggest that the SAC-related genes might be repurposed, it would be interesting to see what happens or which molecular mechanisms are defected upon deletion of those genes (or applying loss of function mutations if possible).

Response to Reviewer #3's comment 3:

We agree with the reviewer that this would be a very interesting worthwhile experiment. Unfortunately, there are no experimental/genetic tools available for *Carpediemonas*, so it is completely impossible to do these experiments. In our initial submission we mentioned that developing *Carpediemonas* as model would be of great importance to increase our understanding on its DNA replication, repair, and cell cycle (initial submission, lines 410-412) but the magnitude of effort required to make this happen should not be underestimated.

Since *Carpediemonas* is not an established model organism, our team will need quite a few highly trained experimentalists over several years to first develop it into an experimentally manipulable model system. This requires the development of:

- 1) A stable and efficient transfection system in *C. membranifera* with selectable markers,
- 2) Methods to knockdown or knockout genes in this organism,
- 3) Reliable immunofluorescence microscopy methods,
- 4) Chromatin immunoprecipitation methods that work in this system, and
- 5) Methods to synchronize cultures and arrest *C. membranifera* in various stages of the cell cycle.

Over the past few years, we have been attempting to develop these approaches for *C. membranifera*, but so far, we have had little success. For instance, co-author Dr. Jerlström-Hultqvist, a technically gifted experimentalist who developed a transfection system for another metamonad (*Spironucleus salmonicida*; Jerlström-Hultqvist *et al* 2012), spent a full year trying, without success, to develop a transfection system in *C. membranifera* (with custom-built plasmid vectors expressing selectable markers and trying multiple different transfection methods including electroporation, chemical means and a variety of animal cell transfection protocols/kits).

Jerlström-Hultqvist *et al* (2012) <https://doi.org/10.1128/EC.00179-12>

Reviewer #3 Comment 4

Overall, this is a very nice genomic analysis study that strongly supports the authors' conclusions. It would be an excellent paper if it was complemented with couple of experiments mentioned above. I also would like to add that I understand that performing experiments might be challenging if the author's expertise (and the teams') is widely computational and not have the required experimental setup in their labs.

Response to Reviewer #3's comment 4:

We thank the reviewer for the careful consideration of our work and their kind assessment of our analyses. We are very interested in all of the scientific questions that the reviewer raises and, for these exact reasons, we have been trying, over several years, to develop *Carpediemonas membranifera* as model. Unfortunately, this turns out to be an extremely challenging task and our attempts have not progressed beyond the early stages, despite extensive investment to date.

In our revision, we have added a paragraph that describes a possible experiment to test our current replication hypothesis, once we have the model system established (lines 443-454). Note that even though we are deeply interested in these questions, the aim of the current manuscript was to investigate the evolutionary diversity of DNA replication, repair, and segregation systems. To address this problem, we completely characterized a novel eukaryotic genome and used state-of-the-art comparative genomics analyses. We believe that our comprehensive analysis has revealed an unprecedented finding that is worthy a standalone publication. While we agree that follow-up experiments in *Carpediemonas* further investigation of the replication and segregation apparatus in this curious organism is important, it must await development of molecular and cell biology tools in the organism. The experimental studies of other metamonads the reviewer has suggested would, in our opinion, depart from the main point of this manuscript.

REVIEWERS' COMMENTS (After manuscript revision)

Reviewer #1 (Remarks to the Author):

The revised version of the manuscript, including extensive additional discussion of analysed systems and improved Figure 1, met all my expectations. All of the points raised in the review were addressed, and I do not have any further comments.

I am excited to see the paper published soon.

Reviewer #2 (Remarks to the Author):

Thank you for addressing my questions and I'm satisfied with the revision.

Reviewer #3 (Remarks to the Author):

I would like to thank the authors for their detailed explanations of why experiments that I suggested are virtually impossible. I totally understand and admit that it is my mistake to treat this species as a model organism just like how it is in eukaryotes. As I briefly stated in my previous report, I understand that experimental approaches are not exactly contributing or possible to a better understanding of a genomics study. I would like to thank the authors again for their careful and detailed explanations for my comments. I have no further questions or requests for this study, I believe the study is complete and ready to be published.

The authors would like to thank all the reviewers for all the comments and questions raised as they helped improve the manuscript. We hope *Carpediemonas membranifera*'s genome assembly becomes a helpful resource for other researchers.